# Coronavirus Infections in Companion Animals: Virology, Epidemiology, Clinical and Pathologic Features

**DOI:** 10.3390/v12091023

**Published:** 2020-09-13

**Authors:** Christine Haake, Sarah Cook, Nicola Pusterla, Brian Murphy

**Affiliations:** 1School of Veterinary Medicine, University of California, Davis, CA 95616, USA; 2Graduate Group Integrative Pathobiology, School of Veterinary Medicine, University of California, Davis, CA 95616, USA; sestevens@ucdavis.edu; 3Department of Medicine & Epidemiology, School of Veterinary Medicine, University of California, Davis, CA 95616, USA; npusterla@ucdavis.edu; 4Department of Pathology, Microbiology, and Immunology, School of Veterinary Medicine, University of California, Davis, CA 95616, USA; bmurphy@ucdavis.edu

**Keywords:** feline infectious peritonitis, coronavirus, canine, ferrets, spike glycoproteins, SARS Virus, COVID-19, zoonoses

## Abstract

Coronaviruses are enveloped RNA viruses capable of causing respiratory, enteric, or systemic diseases in a variety of mammalian hosts that vary in clinical severity from subclinical to fatal. The host range and tissue tropism are largely determined by the coronaviral spike protein, which initiates cellular infection by promoting fusion of the viral and host cell membranes. Companion animal coronaviruses responsible for causing enteric infection include feline enteric coronavirus, ferret enteric coronavirus, canine enteric coronavirus, equine coronavirus, and alpaca enteric coronavirus, while canine respiratory coronavirus and alpaca respiratory coronavirus result in respiratory infection. Ferret systemic coronavirus and feline infectious peritonitis virus, a mutated feline enteric coronavirus, can lead to lethal immuno-inflammatory systemic disease. Recent human viral pandemics, including severe acute respiratory syndrome (SARS), Middle East respiratory syndrome (MERS), and most recently, COVID-19, all thought to originate from bat coronaviruses, demonstrate the zoonotic potential of coronaviruses and their potential to have devastating impacts. A better understanding of the coronaviruses of companion animals, their capacity for cross-species transmission, and the sharing of genetic information may facilitate improved prevention and control strategies for future emerging zoonotic coronaviruses. This article reviews the clinical, epidemiologic, virologic, and pathologic characteristics of nine important coronaviruses of companion animals.

## 1. Introduction

Coronaviruses are spherical, enveloped, single-stranded, positive-sense RNA viruses within the family *Coronaviridae,* named for the ultrastructural “crown-like” (corona) appearance of the spike proteins on the virion surface. Coronaviruses infect humans as well as many other mammalian and avian species, generally causing variably severe intestinal, respiratory, neurologic, or systemic disease syndromes [1,2,3,4]. Genomically, coronaviruses are among the largest of the RNA viruses, with genomes spanning 27.6 to 31 kilobases (kb) in length [5], approximately three times the size of most retroviruses. On the basis of comparative genome sequence analyses, coronaviruses are subdivided into four genera: alphacoronavirus, betacoronavirus, gammacoronavirus, and deltacoronavirus. Alpha- and betacoronaviruses originate from bats and predominantly infect mammals, while gamma- and deltacoronaviruses originate from birds and are capable of infecting both bird and mammal species [6]. Companion animals presently considered include cats, dogs, ferrets, horses, and alpacas. While not universally recognized as companion animals, alpacas and horses are considered by the authors to be companion animals and are therefore included in this review. Notable coronaviruses of companion animals include feline enteric coronavirus (FECV), feline infectious peritonitis virus (FIPV), canine enteric coronavirus (CCoV), ferret enteric coronavirus (FRECV), ferret systemic coronavirus (FRSCV), and alpaca respiratory coronavirus, which are alphacoronaviruses, and canine respiratory coronavirus (CRCoV), equine enteric coronavirus (ECoV), and alpaca enteric coronavirus, which are betacoronaviruses [7]. Phylogenetic relationships of these coronaviruses are shown in Figure 1, while clinical and pathologic features are summarized in Table 1. Other coronaviruses belonging to the betacoronavirus genus include SARS-CoV-1, MERS-CoV, and SARS-CoV-2, zoonotic coronaviruses that have recently transferred from animal to human populations and are capable of causing severe disease and death [8]. The ability of SARS-CoV-2 to initiate infections in companion animals is currently poorly understood, although preliminary studies have indicated that ferrets and cats are permissive for SARS-CoV-2 infection and replication, while the virus has been shown to replicate poorly in dogs, pigs, chickens, and ducks [9]. SARS-CoV-2 infection of horses and camelids has not been reported.

Coronavirus genomes encode three classes of proteins: structural, accessory, and non-structural proteins. Major structural proteins of coronaviruses include the nucleocapsid (N), spike (S), membrane (M), and envelope (E) proteins [5]. The S protein is the primary viral binding protein and mediator of membrane fusion and viral entry. The N protein, in close association with genomic viral RNA (gRNA), forms the helical nucleocapsid, which is stabilized via binding to the M protein (Figure 2). The viral genome and helical nucleocapsid are surrounded by a host-derived lipid bilayer, in which the S, E, and M proteins are anchored. The transmembrane E and M proteins are involved in virion assembly and budding [10]. In addition to the four structural proteins, coronavirus genomes also encode a number of accessory proteins. While the roles of most of the accessory proteins remain poorly understood and may be dispensable for virus replication in vitro, certain accessory proteins appear to enhance viral virulence in vivo; for example, the SARS coronavirus encodes accessory proteins that antagonize the development of type I interferon (IFN) responses [11].

Unlike alphacoronaviruses, a subset of betacoronaviruses are more structurally complex and have additional membrane glycoproteins, called hemagglutinin-esterase (HE) proteins, encoded by an additional gene roughly 1.2 kb in size [12]. Coronavirus HEs are thought to be acquired from an influenza virus C-like gene encoding a hemagglutinin-esterase fusion protein in a relatively recent horizontal gene transfer event [13]. While coronavirus HEs are able to bind to sialic acid, they are reported to serve primarily as receptor-destroying enzymes (RDE), which facilitates the reversibility of the virus-host cell attachment. For all coronaviruses, the S protein is thought to be the primary binding protein, responsible for attachment of coronavirus to the cell surface. However, the contribution of HEs to virion attachment and their role in tissue tropism and pathogenesis are currently not well understood [14].

The molecular events of the coronavirus replication cycle are complex and begin with virion attachment to the host cell, accomplished by binding of the viral S protein to a unique target receptor on the host cell surface. As the primary binding protein and mediator of virus-host cell membrane fusion and subsequent virus entry into the cell, the S protein is critical in determining the host species and tissue and cell tropism for each coronavirus [15]. Upon receptor binding, conformational changes in the S protein expose the fusion peptide, facilitating fusion of the viral and host cell membranes and subsequent release of the viral nucleocapsid into the host cell cytoplasm [10,16]. Upon cytoplasmic release of the viral nucleocapsid, the positive sense genomic RNA (+gRNA) serves as viral messenger RNA (mRNA) for the direct translation of the *replicase* gene complex utilizing the host cell’s ribosomal machinery. The *replicase* gene complex consists of two large open reading frames (ORF) approximately 20 kb in total size [17], ORF1a and ORF1b, the latter transcribed via a ribosomal frameshift. The ORF1a and ORF1b mRNA are translated into polypeptides 1a or 1ab, which are subsequently cleaved by viral proteases to create sixteen nonstructural proteins (nsps). These nonstructural proteins reassemble to form a viral replicase-transcriptase complex, consisting of the RNA-dependent RNA polymerase (RdRp, nsp12), helicase (nsp13), nsps with accessory functions, such as the nsp14 exoribonuclease, as well as multiple membrane-spanning proteins that are thought to provide a membrane-associated scaffold for the assembly of the replicase-transcriptase complex [18,19,20]. As eukaryotic cells typically do not encode an RdRp, that is, they lack the ability to catalyze the formation of RNA using RNA as a substrate, the viral RdRp enzyme provides a useful target for antiviral therapeutics [21,22]. Within this group of nonstructural proteins is an exonuclease with proof-reading function, unusual for RNA viruses but perhaps important for ensuring the fidelity of the very large coronaviral RNA genome during replication [23].

The viral polymerase synthesizes complementary full-length negative-sense RNA copies of the genome, which serve as templates for full length positive-sense RNA genomes, generated via RdRp’s replicase function. In addition to replicase activity, RdRp also has transcriptase activity; by discontinuous RNA synthesis directed by transcriptional regulatory sequences, RdRp creates a set of subgenomic RNAs (sgRNA) of different sizes [24], which are then copied by RdRp into positive-sense mRNAs, serving as templates for translation of viral proteins necessary for virion assembly, including the structural proteins S, E, M, and N. Translated viral proteins are inserted into the cell’s endoplasmic reticulum and then transported to the site of viral assembly, the endoplasmic reticulum-Golgi intermediate compartment (ERGIC). Viral genomes (+gRNA) encapsidated by N proteins bud into the ERGIC membrane, forming fully assembled virions surrounded by a host-derived lipid bilayer [25]. Assembled virions are subsequently transported in vesicles to the plasma membrane, where they are released from the infected cell via exocytosis [26]. In some coronaviruses, the accumulation of S proteins on the surface of infected cells can result in fusion of adjacent cells and the formation of syncytia, facilitating rapid cell-to-cell spread of the virus [27].

The genetic diversity of coronaviruses is a consequence both of polymerase error-driven point mutations, as well as of genetic recombination between different strains and species of coronaviruses during coinfection within the same host cell [5,28]. Relative to other single-stranded RNA viruses, coronavirus mutation rates are moderate to high [29], despite the proof-reading function of the viral exonuclease [30]. Genetic recombination is a direct result of the discontinuous transcriptional activity of the coronaviral polymerase and likely contributes to the emergence of new viruses with altered virulence, novel host species range, and novel tissue tropism [18].

## 2. Feline Enteric Coronavirus and Feline Infectious Peritonitis Virus

### 2.1. Epidemiology, Clinical and Pathologic Features

Feline coronaviruses are separated into two distinct biotypes: feline enteric coronavirus (FECV) and feline infectious peritonitis virus (FIPV). FECV is endemic in domestic cat populations worldwide and primarily infects intestinal enterocytes, typically resulting in either mild enteric disease or a lack of clinical signs (subclinical infections). Experimental studies have demonstrated consistent shedding of FECV in the feces of infected cats from 2 days to 2 weeks post-infection, followed by a decrease in viral loads and intermittent shedding for up to 20 weeks after this period [31,32]. Subclinical carriers of FECV play an important role in shedding and transmitting the virus to other cats via the fecal-oral route, especially in those animals housed indoors in multi-cat environments [33]. FECV primarily infects the apical epithelial cells of the intestinal villi (enterocytes), from the distal duodenum to the cecum. Villous atrophy of the lining mucosa and sloughing and degeneration of epithelial cells at the villous tips occur in severe infections [34]. Shortening and fusion of intestinal villi and hyperplasia of crypt epithelia are also common pathological findings [3].

In contrast to the mild enteric disease or absence of clinical signs associated with FECV infection, the closely related FIPV biotype generally results in a highly inflammatory, systemic, and nearly 100% fatal disease once clinical signs develop. This clinical syndrome is called feline infectious peritonitis (FIP). The origin of FIPV is thought to arise from a select number of spontaneous mutations in the FECV genome, which confers a tropism switch from enterocytes to macrophages, facilitating systemic spread. These mutations are thought to arise *de novo* within each FECV-infected cat. Male cats, purebred cats, and those living in multi-cat environments are more likely to develop FIP [35]. Specific cat breeds at higher risk for the development of FIP include Abyssinian, Bengal, Birmans, ragdoll, and rex cat breeds [36,37], likely due to inherited genetic factors in these breeds, leading to increased susceptibility to FIP [38]. FIP seems to preferentially occur in young, very old, or immunosuppressed individuals. An FIP-like disease has also been documented in a number of wild species of felids infected with coronaviruses, including African lions, cheetahs, mountain lions, leopards, jaguars, lynx, servals, caracal, European wild cats, Sand cats, and Pallas cats [39,40,41,42,43,44,45,46,47].

Key point mutations proposed to be responsible for the conversion of FECV to FIPV include two alternative amino acid differences in the gene encoding the fusion peptide of the spike (S) protein [48], substitutions in the furin cleavage site between receptor-binding (S1) and fusion (S2) domains of the spike protein [49], and mutations in open reading frame 3abc resulting in a truncated protein 3c protein [50]. It has subsequently been reported that one of the amino acid differences in the fusion peptide of the FECV spike protein, specifically, a methionine to leucine substitution at position 1058, is involved in systemic spread of FCEV from the intestine, rather than with the potential to cause FIP [51]. Mutations of 3c and S protein genes are often found in combination, but a single mutation in either S or 3c appears to be sufficient to dramatically alter the tropism of FECV, allowing for enhanced internalization and replication of the virus within monocytes and macrophages, facilitating systemic, cell-associated dissemination of the virus [50].

Clinically, FIP typically manifests as one of two forms: “wet” (effusive) FIP, “dry” (granulomatous) FIP, or some combination of the two. Effusive FIP is the more common and classical form of disease and is generally associated with rapid disease progression and the exudation of fluid into the peritoneal or thoracic body cavities. The “dry”/granulomatous form of FIP generally lacks cavitary effusion and is instead characterized by multifocal granuloma formation in a variety of organs and a more insidious disease progression. As a result, the initial clinical signs of FIP are often nonspecific and may include anorexia, weight loss, and/or chronic fever [52,53]. Clinical neurologic signs, including ataxia, seizures, nystagmus, hyperesthesia, and/or cranial nerve deficits [54,55], as well as ocular disease may occur in some cats, with a higher frequency in those with the dry form of FIP than the wet form [56].

It has been hypothesized that a “strong and focused” cell-mediated immune (CMI) response directed to the coronavirus may prevent FIP disease development, while animals with a “weak” CMI in combination with a strong humoral immune response will likely develop wet FIP. Further, it has been hypothesized that animals with a “moderate” CMI will likely develop the dry form of FIP [57].

Grossly, the wet/effusive form of FIP is characterized by “straw-colored,” semi-translucent, protein-rich peritoneal or thoracic effusions and fibrinous and granulomatous serositis/pleuritis with variable involvement of parenchymal organs (Figure 3A) [56]. Virus-associated pyogranulomatous inflammation is focused on small and medium sized veins, resulting in vascular injury and leakage [58]. These vasculo-centric lesions can occur in the omentum and serosal surfaces of the liver, spleen, intestines, kidneys, and lungs, and are composed primarily of macrophage aggregates in combination with smaller numbers of neutrophils and lymphocytes (pyogranulomatous inflammation). Occasionally, pyogranulomatous nodular lesions extend beyond the serosal surfaces into underlying parenchyma. Coronaviral antigen can often be detected within intralesional macrophages using immunohistochemistry techniques; coronaviral antigen detection via immunohistochemistry is a commonly utilized diagnostic method.

The dry form of FIP is characterized grossly by variably sized parenchymal and serosal pyogranulomas in affected organs but lacks the exudation archetypal of wet FIP. Granulomatous lesions of dry FIP may extend from serosal surfaces into the parenchyma of affected organs, and lesions may be restricted to a single organ, such as the kidney, eye, or brain [59]. Other frequently affected organs in the dry form of FIP include the mesenteric and mediastinal lymph nodes, omentum, intestine, and liver. Perivascular inflammatory lesions contain aggregates of macrophages and fewer neutrophils, which are surrounded by dense infiltrates of primarily B lymphocytes and plasma cells extending into surrounding tissues, with or without the presence of vasculitis [47,59]. It is not uncommon for affected animals to demonstrate some combination of both effusive and granulomatous forms of the disease.

It has been hypothesized that immune-mediated type III and/or type IV hypersensitivity reactions may play a role in the perivascular granulomatous inflammation characteristic of FIP [60,61]. Type III hypersensitivity lesions feature the overproduction of immune complexes comprised of antibody bound to soluble antigen, and subsequent inflammatory pathway activation, deposition into vessel walls (e.g., vasculitis) and tissue injury [62]. FIPV-associated vascular lesions have been hypothesized as being caused by type III hypersensitivity based on studies demonstrating antibody, complement, and FCoV antigen within vascular lesions; however, a definitive connection between type III hypersensitivity and FIP-associated vasculitis/peri-vasculitis has not been unequivocally confirmed [60,63]. Alternatively, FIPV-associated vascular injury and subsequent permeability may be a result of virus-induced activation of monocytes and macrophages. This hypothesis is supported by the finding that vascular endothelial growth factor (VEGF) produced by FIPV-infected monocytes and macrophages causes vascular permeability and effusion in cats with FIP [64]. Matrix metalloprotease 9 (MMP-9) has also been shown to be upregulated in activated monocytes and macrophages in FIP, contributing to the destruction of type IV collagen and degradation of the basal lamina of affected vessels in FIP vasculitis [58]. Type IV, or delayed-type hypersensitivity, is mediated by hyperstimulated T cells and macrophages, which cause damage to surrounding tissue and may contribute to the granuloma formation characteristic of the dry form of FIP [65].

In a subset of wet and dry FIP cases, affected cats may present with ocular and/or neurologic involvement. In cases with ocular involvement, diffuse and multifocal inflammatory infiltrates may be found in the ciliary body, retina, and choroid as well as throughout the uvea and in the sclera, conjunctiva, and optic nerve. Ocular perivascular leukocytes are typically lymphoplasmacytic, composed mostly of B cells and plasma cells, with fewer numbers of T cells and macrophages [66]. Gross lesions of FIP in the central nervous system include ventricular dilation, flattening of cerebral gyri, and ependymal and meningeal congestion [67]. Mild to marked ventricular enlargement is associated with accumulation of inflammatory cells within the ventricles [54], with corresponding increased protein, increased cellularity, and presence of virus in cerebrospinal fluid [67]. Histopathological lesions in the brain consist of perivascular neutrophilic and lymphoplasmacytic infiltrates in the leptomeninges, the choroid plexus, the periventricular space, and/or the parenchyma of the spinal cord and brainstem [68]. Less common (atypical) manifestations of FIP include nodular dermatitis [69,70,71], rhinitis [72], orchitis [73,74], priapism [75], and syringomyelia associated with involvement of the fourth ventricle [76].

### 2.2. Virology

Feline coronaviruses are alphacoronaviruses and are divided into two serotypes, type I and type II, based on genetic and antigenic properties [77]. Although both serotypes are capable of causing FIP [78], serotype I is much more prevalent in nature and is responsible for 80–90% of naturally occurring clinical cases [79,80]. Serotype II is comparatively rare, having emerged as a result of recombination events between feline coronavirus serotype I and canine enteric coronavirus serotype II, following cross-species transmission of CCoV to cats [81,82,83]. Although the feline serotype I is more prevalent, it is less well studied due to challenges in propagating this viral serotype in culture-adapted cell lines in vitro. The molecular events of the serotype II viral lifecycle are better understood due to the relative ease with which serotype II can be propagated and studied in vitro.

The target cell membrane receptor of serotype II feline coronaviruses has been identified as feline aminopeptidase N (fAPN) [84]. Feline APN is a membrane peptidase expressed on the brush border of small intestine and renal tubule microvilli, as well as by cells of myeloid origin, including monocytes, macrophages, and granulocytes [85]. Additional non-specific viral receptors, including the lectin molecule DC-SIGN, have also been proposed [41,78]. The primary cell receptor for the more prevalent serotype I FIPV has yet to be definitively identified [86]; however, studies have proposed DC-SIGN [87], as a potential host cell entry co-receptor. The Fc receptor CD16 (FcyRIII) has also been proposed as a potential cellular receptor [88]. Interestingly, antibodies directed to the spike protein of feline coronavirus have been shown to enhance virus infection both in vitro [89] and in vivo [63] through a mechanism known as antibody dependent enhancement (ADE). In ADE, antibodies facilitate the uptake of virus-antibody complexes by monocytes and macrophages using Fc receptors like CD16, resulting in more efficient infection than by virus alone [90,91].

Following binding to the target receptor on the cell surface, FIPV serotype II enters monocytes via clathrin and caveolae-independent and dynamin-dependent endocytosis [92]. Consistent with the initial distribution of lesions on serosal surfaces of abdominal organs, FIPV is thought to preferentially target peritoneal macrophages [88]. Once ensconced within these histiocytic cells, FIPV is able to seed the abdominal and thoracic cavities and, in some cases, spread to more distant sites, such as the brain and eye.

## 3. Ferret Enteric Coronavirus and Ferret Systemic Coronavirus

### Epidemiology, Virology, and Clinical and Pathologic Features

Similar to feline coronaviruses, infection with ferret coronaviruses can result in enteric or systemic disease [93]. First identified in the United States in 2000, ferret enteric coronavirus (FRECV) is associated with epizootic catarrhal enteritis (ECE), originally called “green slime disease” due to the development of profuse, foul-smelling, bright green mucus-laden diarrhea [94]. Clinically, ECE is associated with lethargy, anorexia, and vomiting. ECE is characterized by high morbidity but low mortality [95]. While juvenile ferrets develop mild to subclinical disease and can be subclinical carriers, ECE can cause more severe disease in older ferrets [96].

A genetically distinct coronavirus called ferret systemic coronavirus infection (FRSCV) was subsequently identified, which causes a systemic, progressive, and fatal pyogranulomatous inflammatory disease resembling the dry form of feline infectious peritonitis (FIP) in cats. The average age at the time of FRSCV diagnosis has been reported to be 11 months, and clinical signs include chronic weight loss, anorexia, diarrhea, palpable abdominal masses, and neurologic disease [97].

Both the enteric (FRECV) and systemic (FRSCV) ferret coronaviruses are alphacoronaviruses, related to feline coronavirus and canine enteric coronavirus, and most closely related to mink coronavirus. Complete genome sequencing of FRSCV and FRECV strains revealed a shared 89% nucleotide identity, but only 49.9–68.9% nucleotide identity with other known coronaviruses [98]. The pathogenic relationship of these two ferret coronaviruses, and whether FRSCV arises by mutation within ferrets infected with FRECV, has not been determined. The cellular entry receptors have also not been identified. Many facets of the pathogenesis of the virulent systemic ferret coronavirus remain unknown, but as is true for FIPV, macrophages appear to play an important role in the inflammatory response. Furthermore, similarities in pathologic lesions suggest parallels in the pathogenesis of ferret systemic coronavirus and FIP.

FRECV is associated with lesions restricted to the gastrointestinal tract, which include lymphocytic enteritis, villous blunting, fusion, and atrophy, as well as vacuolar degeneration and necrosis of apical villous enterocytes, similar to FECV [94].

In contrast to FRECV, FRSCV is grossly associated with pale to white nodules (granulomatous inflammation) in multiple organs, including the spleen, kidneys, mesenteric lymph nodes, intestines, liver, lungs, and brain (Figure 4) [99,100]. Granulomas have a heterogenous cellular composition including macrophages, T and B lymphocytes, and plasma cells. These granulomatous lesions are morphologically similar to FIP both in immune cell composition and presence of virus within the macrophage cytoplasm [101]. Interestingly, cavitary effusions and vasculitis, characteristic features of the wet form of FIP, have not been identified in the majority of ferrets infected with systemic coronavirus. Clearly, much remains to be learned about ferret coronavirus virology and pathology.

## 4. Canine Enteric Coronavirus

### 4.1. Epidemiology and Clinical Features

As is true of FECV, canine enteric coronavirus (CCoV) is a common infection of dogs with worldwide distribution. While not universally recognized as an important canine enteric pathogen, multiple independent studies have demonstrated that CCoV is significantly associated with diarrhea in dogs [102,103]. CCoV is transmitted via the fecal-oral route, with higher prevalence in dogs housed in dense populations such as in shelters or kennels [104]. First reported in 1971 in dogs in a canine military unit in Germany [105], CCoV generally causes mild, self-limiting diarrhea in dogs, especially in young puppies. More severe hemorrhagic disease associated with higher mortality has also been reported in combination with other pathogens [106], including canine parvovirus type 2 [107] and canine adenovirus type I [108]. CCoV infection has a synergistic effect with canine parvovirus type 2, increasing severity of enteric disease [109]. More virulent strains of CCoV, capable of causing significant enteric disease in the absence of coinfection have recently been reported [110], as well as pantropic strains that cause a fatal systemic disease involving lethargy, inappetence, vomiting, hemorrhagic diarrhea, ataxia, and seizures [111,112,113]. CCoV has also been detected in a number of wild canids, including foxes and raccoon dogs in China [114] and wolves in Alaska [115] and Europe. Remarkably, sequences of the CCoVs found in European wolves were up to 98–99% homologous to known CCoV sequences isolated from domestic dogs [116].

### 4.2. Virology

Similar to FECV, two serotypes of the CCoV exist: serotypes I and II. Mixed infections with strains of both serotypes are common [117]. Like the feline coronavirus serotype II, CCoV serotype II strains replicate well in tissue culture and use APN as an entry receptor. The cellular receptor for serotype I viruses has not been determined, as these viruses are much more difficult to propagate in tissue culture systems. CCoV serotypes I and II share close to 96% nucleotide identity throughout most of their genome, while the gene encoding the S protein is much more divergent, with only 56% sequence identity. It is likely that FECV serotype I and CCoV serotype I arose from a common viral ancestor, while CCoV serotype II arose via recombination with an unknown coronavirus, in the process acquiring an antigenically distinct S gene [83].

The continuing evolution of canine enteric coronaviruses with altered virulence and tropism is likely a result of changes in the genome due to random point mutations and periodic genetic recombination. Genetic recombination between serotype II CCoV and other coronaviruses resulted in the emergence of canine coronavirus variants with spike protein N-terminal domains that are largely homologous to transmissible gastroenteritis virus (TGEV), a coronavirus of pigs [118].

### 4.3. Pathology

Similar to the pathology of other enteric coronaviruses, CCoV infects and replicates in the apical and lateral enterocytes of the intestinal villi (mature enterocytes), resulting in cellular degeneration and/or necrosis characterized by atrophy of enterocytes, loss of the cellular brush border, and sloughing of necrotic cells into the intestinal lumen. Degeneration and destruction of mature enterocytes at the villous tips can lead to villous atrophy, ultimately resulting clinically in maldigestion, malabsorption and diarrhea [119].

A more severe form of enteritis in puppies infected with CCoV has also been reported, in the absence of co-infection. Gross pathology in one case revealed moderate, diffuse, hemorrhagic enteritis, and in another, severe ileo-cecal intussusception and segmental necrotic enteritis. Histologically, mild, lymphocytic and plasmacytic enteritis was present in the first case, along with necrosis and enteric and splenic lymphoid depletion. In the second case, depletion of gut associated lymphoid tissues was also noted, along with diffuse villous blunting and crypt necrosis [110]. A case report of pantropic CCoV described lesions in multiple organs, including a fibrinopurulent bronchopneumonia, renal cortical infarcts, severe coalescing centrilobular hepatic fatty change, and multifocal hemorrhage in the spleen with lymphoid depletion. Chronic diffuse enteritis in this case was associated with the presence of adult ascarids in addition to CCoV [120].

## 5. Canine Respiratory Coronavirus

### 5.1. Epidemiology and Clinical Features

First discovered in 2003 in dogs housed at a rehoming kennel in the United Kingdom [121], canine respiratory coronavirus (CRCoV) is a coronavirus with worldwide distribution [122,123,124,125] and a significant etiologic component of canine infectious respiratory disease (CIRD) or “kennel cough” [126]. CIRD is a highly contagious, polymicrobial respiratory disease syndrome associated with a number of bacterial and viral agents and is readily transmitted via aerosols between dogs housed in relatively high-density groups, like shelters or kennels. Pathogens associated with CIRD include CRCoV, canine adenovirus 2 (CAV-2), canine parainfluenza virus (CPIV), *Bordetella bronchiseptica*, canine herpesvirus, canine pneumovirus (CnPnV), *Streptococcus equi* subsp. *zooepidemicus,* and *Mycoplasma* spp [127,128]; infection by one or a combination of these pathogens may result in disease.

CRCoV has also been shown to be capable of causing disease on its own [4] and is thought to play a role in early CIRD infection by damaging the mucociliary elevator. Affected dogs have an impaired ability to clear pathogens and foreign material from the lower respiratory tract, predisposing them to secondary infections and more severe clinical disease [129]. CRCoV is spread by aerosol transmission and is most commonly associated with mild signs of upper respiratory disease, including nasal discharge, sneezing, and coughing [4]. As is true of SARS CoV-2, CRCoV can also be associated with more severe clinical signs, inappetence, and bronchopneumonia. Disease occurs most frequently in fall to winter months [130], and populations most at risk are dogs densely housed in shelter, kennel, or group environments [131]. CRCoV has been proposed as a naturally occurring animal model of SARS-CoV-2 infection in humans, due to parallels in pathogenesis and early host immune response [132].

### 5.2. Virology

CRCoV is a betacoronavirus, genetically distinct from the alphacoronavirus, canine enteric coronavirus. Based on the polymerase gene sequence, the two canine coronaviruses have sequence identity of 68.5%, but only 21.1% similarity based on the Spike gene [123]. Among betacoronaviruses, CRCoV has the highest sequence identity with the polymerase gene of bovine coronavirus (BCoV) (98.8%) and human “common cold” coronavirus OC43 (98.4%), all of which cause mild to moderate upper respiratory disease in their respective hosts [126]. As is true for other betacoronaviruses, CRCoV binds initially to sialic acids and heparan sulfate on the cell surface for attachment, prior to cell entry via caveolin-dependent endocytosis [133]. Human leukocyte antigen class I (HLA-1), a human transmembrane glycoprotein, has been shown to act as the entry receptor for the in vitro infection of human airway epithelial cells by both CRCoV and BCoV [134].

### 5.3. Pathology

Histopathological lesions are most significant in the trachea and nasal cavity, where infection with CRCoV causes inflammation and damage to the ciliated respiratory epithelium, impairing the clearance of particulate matter in the lower respiratory tract and predisposing individuals to secondary bacterial infection of the lungs. Histological examination following experimental infection with CRCoV has demonstrated that the epithelia of the respiratory tract is disordered and devoid of cilia and goblet cells, and inflammatory cells infiltrate within the epithelium and subjacent lamina propria [4]. The trachea and nasal tonsil are the most common sites of CRCoV infection and are reported to have the highest viral loads, detected by quantitative RT-PCR. Though infrequent, CRCoV has also been detected in the spleen, mesenteric lymph node, and colon of infected dogs; while this may indicate the potential of CRCoV to display a dual tropism, it is likely that the detection of CRCoV outside the respiratory tract is a result of passive transport from the respiratory tract through the ingestion of saliva and respiratory secretions. [135].

## 6. Equine and Alpaca Coronaviruses

### 6.1. Epidemiology and Clinical Features

Equine coronavirus (ECoV) is an enteric coronavirus originally reported in 2000 in a neonatal foal with enterocolitis [136]. Sporadic outbreaks in riding, racing, and show horses have been reported in the USA, Europe, and Japan with increasing frequency. Clinically, ECoV is associated with anorexia, fever, and lethargy, and in some cases, diarrhea, colic, and neurologic signs [137]. While ECoV infections are generally self-limiting, severe damage to the intestinal mucosa and subsequent loss of barrier function can lead to mortality due to secondary endotoxemia, septicemia, and hyperammonemia-associated encephalopathy [138]. Signs of encephalopathy associated with ECoV infection have been reported in 3% of clinical cases and include circling, head pressing, ataxia, proprioceptive deficits, nystagmus, recumbency, and seizures [138,139]. Hyperammonemia may be caused by increased ammonia production due to enteric microbiome dysbiosis associated with ECoV infection or increased absorption of ammonia from the gastrointestinal tract due to breakdown of the normal intestinal mucosal barrier function [137]. Similar to CRCoV, infections appear to be increased during colder months. While mainly affecting adult horses, infection in foals is associated with more severe gastrointestinal disease. Transmission is fecal-oral [140,141], and it is likely that subclinical horses play a role in transmission of the virus [142].

Alpaca enteric coronavirus is associated with outbreaks of diarrhea in llamas and alpacas; an Oregon study found alpaca enteric coronavirus to be the most common pathogen causing diarrhea in unweaned crias. Alpaca enteric coronavirus was noted to cause diarrhea throughout the year and was involved in outbreaks affecting adult animals as well as unweaned crias ranging in age from 1 to 7 months old [143].

A strong epidemiologic association has been made between alpaca respiratory coronavirus and an outbreak of alpaca respiratory syndrome (ARS) in alpacas in 2007. ARS is characterized by acute respiratory signs ranging in severity from mild upper respiratory disease to severe respiratory distress, high fever, and death [144]. Though all signalments can be affected, ARS is primarily reported in pregnant alpacas; severe fetal hypoxia in alpacas with ARS can result in abortion [145].

### 6.2. Virology

Equine coronavirus is a betacoronavirus, classified in the same genus as canine respiratory coronavirus. The cellular entry receptor has not been identified. Complete genome sequences have been determined for three ECoV isolates from Japan and one from the USA [146,147]. All three isolates from Japan were genetically similar to the isolate from the USA (NC99), with a sequence identity between 98.2 to 98.7% [147]. ECoV is phylogenetically related to a wide variety of coronaviruses including bovine coronavirus, human coronavirus OC43, and porcine hemagglutinating encephalomyelitis virus. Compared to these three coronaviruses, the ECoV nsp3 protein, a critical component of the replicase-transcriptase complex, is the most divergent, containing 3 amino acid deletions and 55 amino acid insertions, though the functional significance of these insertions and deletions has not been clarified [146].

Similar to equine coronavirus, the enteric alpaca coronavirus is a betacoronavirus, first recognized as causing severe diarrhea in llamas and alpacas in 1998 [148]. The enteric alpaca coronavirus is most closely related to bovine coronavirus (>99.5% nucleotide identity), human coronavirus OC43 (>96% identity), and porcine hemagglutinating encephalomyelitis virus (>93% identity), with the most significant differences present in the spike protein sequences [149]. CRCoV, ECoV, and alpaca betacoronavirus are all either thought to descend from BCoV or have a common ancestor, likely a rat betacoronavirus [7,150].

A novel alpaca coronavirus belonging to the alphacoronavirus genus was isolated in 2007 and associated with acute respiratory disease rather than enteric disease [144]. Complete genome sequencing revealed less than 50% nucleotide identity with the previously reported enteric alpaca coronavirus but a much higher 92.2% nucleotide identity with the human coronavirus (HCoV) 229E, with striking similarity between the HCoV 229E and alpaca respiratory coronavirus spike proteins. Comparison of spike gene sequences revealed that alpaca respiratory coronavirus is most similar to HCoV 229E strains isolated between the 1960s and 1980s, suggesting the possibility that a transmission event may have occurred between alpacas and humans [151].

### 6.3. Pathology

The pathology of equine coronavirus in two horses and one donkey ranging in age from 6 months to 11 years old has been described. The naturally infected equids had severe diffuse necrotizing enteritis characterized by marked villous attenuation, necrosis of apical enterocytes in the small intestinal villi, pseudomembrane formation, and hemorrhage and microthrombosis within the mucosa and submucosa (Figure 5B). In contrast to enteric coronavirus infections in the carnivores, ECoV infection in horses has also been associated with crypt necrosis. In cases of hyperammonemia-associated encephalopathy, Alzheimer type II astrocyte hypertrophy and hyperplasia were observed diffusely throughout the cerebral cortex [2].

Gross pathology has been reported in one adult alpaca infected with enteric alpaca coronavirus; gross findings included diffuse thickening of the wall of the third gastric compartment, enlarged dark red mesenteric lymph nodes, and watery intestinal contents mixed with mucous. Histopathological lesions in the small intestine included moderate diffuse edema of the lamina propria and submucosa, multifocal petechiae of the mucosa and submucosa in several sections, moderate autolysis, and small amounts of necrotic debris present within crypts. Mesenteric lymph node sinusoids were hemorrhagic, with fibrinopurulent exudate present in the lymph node parenchyma and lymphatics. Nutritional stress, copper, and other mineral deficiencies may have played a role in the severity of disease seen in this 4-year old alpaca [148].

Alpaca respiratory coronavirus pathology has been reported in 11 naturally infected alpacas: Gross findings included severe pulmonary congestion and edema in combination with marked pleural effusion. Histologically, pulmonary congestion and edema were associated with marked, diffuse interstitial to bronchointerstitial pneumonia focused on terminal airways, including intraluminal fibrin deposition and hyaline membrane formation. Epithelial necrosis and regenerative hyperplasia between terminal airways and alveolar ducts and macrophage infiltrates within the septa and lumen were also noted in several cases [144].

## 7. Zoonotic Coronaviruses

Coronaviridae is a highly successful family of viruses that infects many different vertebrate classes and orders, including humans, causing diseases that range from localized respiratory or enteric infections to systemic disease. Coronaviruses cause significant morbidity and mortality in companion and agricultural animal species, including dogs, cats, ferrets, horses, alpacas, pigs, bovids, and poultry, as well as numerous species of wild animals. Cross-species transmission of canine enteric coronavirus to cats, leading to genetic recombination between FECV serotype I and CCoV serotype II, resulted in the emergence of FECV serotype II [81]. Similarly, recombination between serotype II CCoV and other coronaviruses resulted in the emergence of canine coronavirus variants with spike protein N-terminal domains that are largely homologous to transmissible gastroenteritis virus (TGEV), a coronavirus of pigs [118]. Viral polymerase error-driven point mutations, genetic recombination between different strains and species of coronaviruses, and the incorporation of genes from other viral taxa via nonhomologous recombination [152] demonstrate the genetic plasticity of coronaviruses and contribute to the alarming ability of coronaviruses to “jump species” [153]. Over the past 20 years, three human coronaviral pandemics (SARS, MERS, and most recently, COVID-19) all thought to originate from bat coronaviruses [8], demonstrate the zoonotic potential of coronaviruses. Progressive pathogen emergence along with amplified rates of global dissemination [154] represent a profound threat to global health and world economies. In alignment with the concept of “One Health,” a more thorough understanding of the coronaviruses of companion animals, their biological properties, their ability to recombine and to acquire new biological attributes, and their capacity for cross-species transmission has the potential to improve prevention and control measures for future emerging zoonotic coronaviruses [155].

## Figures and Tables

**Figure 1 viruses-12-01023-f001:**
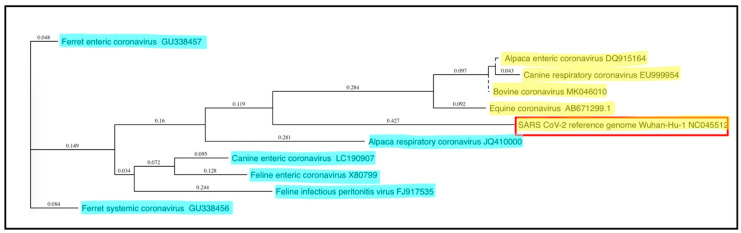
Phylogenetic relationships of coronaviruses of companion animals. The 3′ portions of the coronaviral genomes encoding the spike and other non-structural proteins (~9 kb) were compared and plotted as a “guide tree” using MacVector software (ClustalW Multiple Sequence Alignment). Betacoronavirus sequences are highlighted in yellow, while alphacoronavirus sequences are highlighted in blue; the zoonotic SARS CoV-2 coronavirus is surrounded by a red box. GenBank submission numbers are indicated for each sequence.

**Figure 2 viruses-12-01023-f002:**
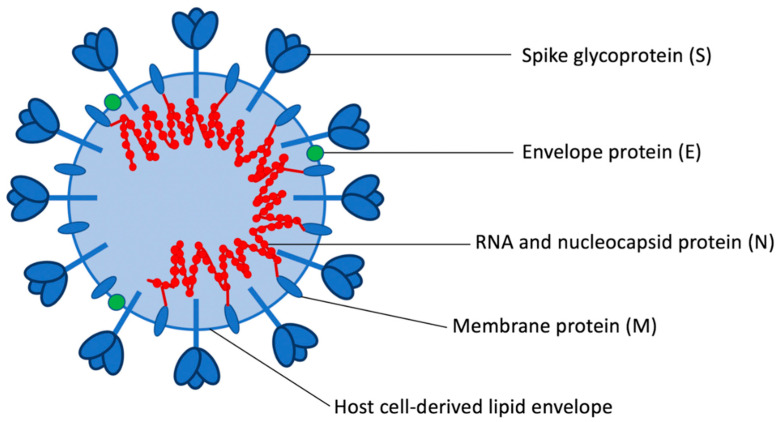
Coronavirus structural proteins.

**Figure 3 viruses-12-01023-f003:**
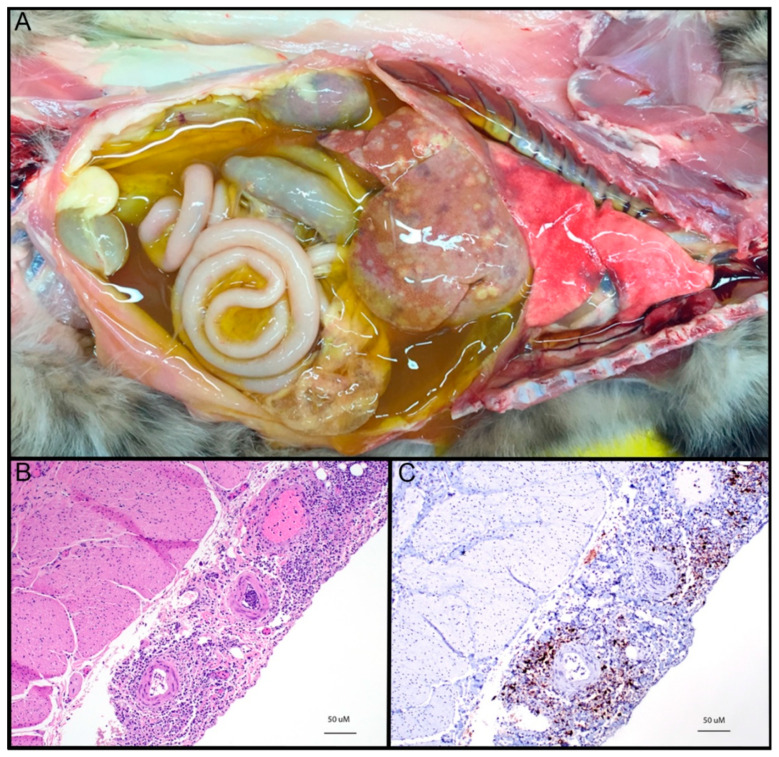
(**A**) Gross image of “wet” or effusive feline infectious peritonitis (FIP), thoracic and abdominal cavities, cat. Abundant semi-translucent “straw-colored”, proteinaceous peritoneal effusion with fibrinous and granulomatous serositis and multifocal granulomatous lesions in the liver. Gross image courtesy of Chrissy Eckstrand. (**B**) FIP, urinary bladder serosal surface, cat, hematoxylin and eosin (HE). Severe, necrotizing, pyogranulomatous and lymphoplasmacytic serositis and vasculitis. (**C**) FIP, urinary bladder serosal surface, cat, FCoV immunohistochemistry. Same lesion tissue as 3b with frequent, positive immunoreactivity for FCoV antigen (brown pigment).

**Figure 4 viruses-12-01023-f004:**
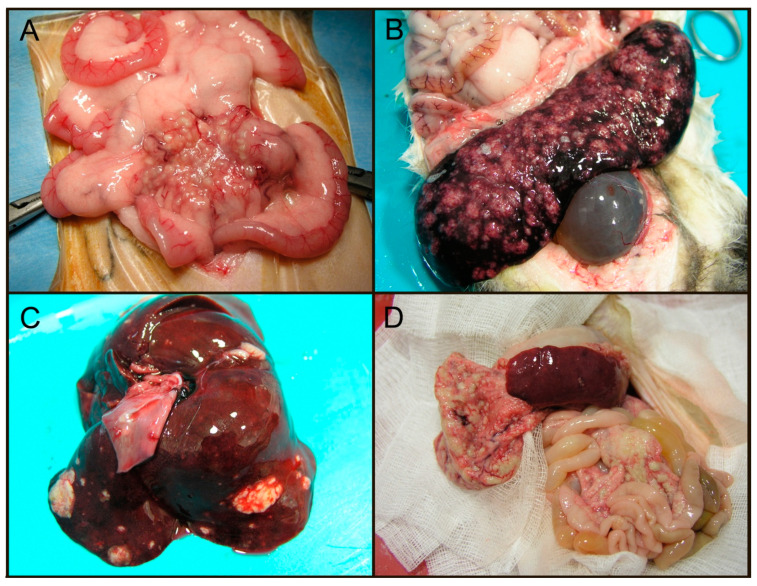
Gross lesions associated with ferret systemic coronavirus (FRSCV). (**A**) Ferret, coronavirus-associated granulomatous mesenteritis. Numerous, multifocal to coalescing, pale tan, firm nodular masses (granulomas) distributed throughout the mesentery, often corresponding to vasculature. (**B**) Ferret, coronavirus-associated serositis and splenitis. Numerous, multifocal to coalescing, pale tan nodules (granulomas) expanding the serosa with variable parenchymal involvement. (**C**) Ferret, coronavirus-associated hepatitis. Multifocal, pale tan, expansile nodular masses throughout the liver. (**D**) Ferret, coronavirus-associated peritonitis. Multifocal to coalescing, pale tan, nodular masses (granulomas) throughout the peritoneum. All images courtesy of Jordi Jimenez.

**Figure 5 viruses-12-01023-f005:**
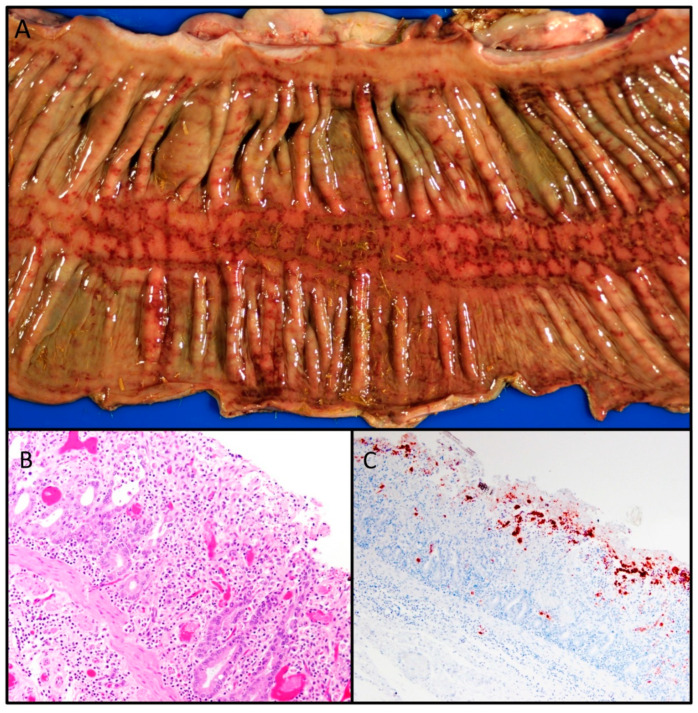
(**A**) Equine coronavirus-associated colitis, colon, horse. Moderate, necrohemorrhagic colitis. Image courtesy of Silvia Siso. (**B**) Equine coronavirus-associated enteritis, jejunum, horse. Mixed inflammatory enteritis with crypt ectasia and necrosis (crypt “abscesses”) and microvascular thrombi. (**C**) Equine coronavirus-associated enteritis, jejunum, horse. Diffuse immunoreactivity at the tips of necrotic villi using bovine coronavirus antiserum (immunohistochemistry). Figure 5B,C courtesy of Federico Giannitti.

**Table 1 viruses-12-01023-t001:** Clinical and pathologic features of major coronavirus infections of companion animals.

Virus	Primary Target Organ(s)	Primary Host Cell	Disease/Symptoms	Transmission	Cellular Receptor
Genus Alphacoronavirus					
Feline enteric coronavirus (FECV)	GI tract	Enterocyte	Asymptomatic to mild gastroenteritis and diarrhea	Direct contact; fecal-oral, maternal shedding	Serotype I: unknown Serotype II: APN
Feline infectious peritonitis virus (FIPV)	Omentum, serosal/pleural surfaces, liver, kidneys, lymph nodes, eyes, brain	Monocyte, macrophage	Peritonitis, thoracic and abdominal effusions, CNS and ocular signs.	Rare to no horizontal transmission	Serotype I: unknown Serotype II: APN
Ferret enteric coronavirus (FRECV)	GI tract	Enterocyte	Epizootic catarrhal enteritis	Fecal-oral	Unknown
Ferret systemic coronavirus (FRSCV)	Spleen, mesenteric lymph nodes, intestines, kidneys, liver, lungs, brain	Unknown	Weight loss, anorexia, diarrhea, abdominal granulomas/masses, CNS signs	Unknown	Unknown
Canine enteric coronavirus (CECoV)	GI tract	Enterocyte	Mild gastroenteritis and diarrhea; rarely, severe enteritis and systemic signs	Fecal-oral	Serotype I: unknown Serotype II: APN
Alpaca respiratory coronavirus	Respiratory tract	Respiratory epithelia (presumed)	Mild to severe respiratory disease	Aerosol (presumed)	Unknown
Genus Betacoronavirus					
Canine respiratory coronavirus (CRCoV)	Respiratory tract	Respiratory epithelia	Mild upper respiratory disease	Aerosol	Unknown
Equine coronavirus (ECoV)	GI tract	Enterocyte	Fever, anorexia, lethargy; less frequently, diarrhea, colic, neurologic signs	Fecal-oral	Unknown
Alpaca enteric coronavirus	GI tract	Enterocyte (presumed)	Enteritis, severe diarrhea	Fecal-oral (presumed)	Unknown

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
