# Peer review of "Coronavirus Infections in Companion Animals: Virology, Epidemiology, Clinical and Pathologic Features"

_viruses, 2020, doi:10.3390/v12091023_

Round 1

Reviewer 1 Report

Companion animal: The definition differs depending on the country or region. I personally find it strange that alpaca and horses are companion animals. Definitions for these animals should be included in the introduction.

Author Response

Point 1. Companion animal: The definition differs depending on the country or region. I personally find it strange that alpaca and horses are companion animals. Definitions for these animals should be included in the introduction.

Response 1. The manuscript has been revised to include a definition of the companion animals considered in this review, while recognizing that horses and alpacas are not universally recognized as companion animals.

Reviewer 2 Report

This is a comprehensive review of the existing literature about coronaviruses in companion animals. Although other authors have addressed this topic after the emergence of SARS-CoV-2 pandemic, this review appears to be well written and organized and could be published after addressing some issues and update the reference list. As a main suggestion, a phylogenetic tree showing how these coronaviruses are correlated each to other and to other animal coronaviruses is required.

Line 50. add the following ref: N. Decaro, A. Lorusso. Novel human coronavirus (SARS-CoV-2): a lesson from animal coronaviruses Vet. Microbiol. (2020) https://doi.org/10.1016/j.vetmic.2020.108693

Lines 161-168. The mutations in the S protein have been proven to be involved in the systemic spread of the virus, which not always associated to occurrence of FIP (Porter E, Tasker S, Day MJ, et al. Amino acid changes in the spike protein of feline coronavirus correlate with systemic spread of virus from the intestine and not with feline infectious peritonitis. Vet Res. 2014;45(1):49. Published 2014 Apr 25. doi:10.1186/1297-9716-45-49)

Line 168: citation missing.

Lines 212-226. The role of metalloproteases secreted by infected monocytes/marcophages in the pathogenesis of FIP should be mentioned.

Line 247. add the following ref:Lorusso A, Decaro N, Schellen P, et al. Gain, preservation, and loss of a group 1a coronavirus accessory glycoprotein. J Virol. 2008;82(20):10312-10317. doi:10.1128/JVI.01031-08.

Line 297. Use acronyms.

Line 312. Citation missing. Cite Decaro N, Buonavoglia C. An update on canine coronaviruses: viral evolution and pathobiology. Vet Microbiol. 2008 Dec 10;132(3-4):221-34. doi: 10.1016/j.vetmic.2008.06.007.

Line 319. Add the following ref: Alfano F, Dowgier G, Valentino MP, et al. Identification of Pantropic Canine Coronavirus in a Wolf ( Canis lupus italicus) in Italy. J Wildl Dis. 2019;55(2):504-508. doi:10.7589/2018-07-182.

Lines 308-314. The authors should discuss that CCoV is not universially recognized as an important canine enteric pathogen, although two independent studies demonstrated that this virus is significantly associated to the occurrence of diarrhoea in dogs (Dowgier G, Lorusso E, Decaro N, et al. A molecular survey for selected viral enteropathogens revealed a limited role of Canine circovirus in the development of canine acute gastroenteritis. Vet Microbiol. 2017;204:54-58. doi:10.1016/j.vetmic.2017.04.007; Duijvestijn M, Mughini-Gras L, Schuurman N, Schijf W, Wagenaar JA, Egberink H. Enteropathogen infections in canine puppies: (Co-)occurrence, clinical relevance and risk factors. Vet Microbiol. 2016;195:115-122. doi:10.1016/j.vetmic.2016.09.006).

Lines 361-364. Emerging pathogens have been also associated to CIRD, including canine pneumovirus and Streptococcus equi subsp, zooepidemicus (Decaro N, Mari V, Larocca V, et al. Molecular surveillance of traditional and emerging pathogens associated with canine infectious respiratory disease. Vet Microbiol. 2016;192:21-25. doi:10.1016/j.vetmic.2016.06.009).

Lines 395-397. Detection of CRCoV in the canine enteric tract does not mean that the virus has a dual tropism, since most likely this is related not to active replication in the gut but to passive transport from the respiratory tract through ingestion of saliva and respiratory secretions. Please, amend the sentence.

Line 411. Use the acronym.

Paragraph 6.1. No data are reported about epidemiology and clinical features of alpaca coronavirus.

Paragraphs 5.2 and 6.2. the authors should mention that CRCoV, ECoV and alpaca betacoronavirus are strictly related to BCoV since they descend from the bovine virus or alternatively all these viruses have a common ancenstor, likely a rat betacoronavirus (see Decaro and Lorusso, 2020).

Paragraph 6.3. Nothing about alpaca coronavirus pathology?

Author Response

Point 1. This is a comprehensive review of the existing literature about coronaviruses in companion animals. Although other authors have addressed this topic after the emergence of SARS-CoV-2 pandemic, this review appears to be well written and organized and could be published after addressing some issues and update the reference list. As a main suggestion, a phylogenetic tree showing how these coronaviruses are correlated each to other and to other animal coronaviruses is required.

Response 1. The manuscript has been revised to include a phylogenetic tree demonstrating the genetic relationship of each coronavirus to the other animal coronaviruses.

Point 2. Line 50. add the following ref: N. Decaro, A. Lorusso. Novel human coronavirus (SARS-CoV-2): a lesson from animal coronaviruses Vet. Microbiol. (2020) https://doi.org/10.1016/j.vetmic.2020.108693

Response 2. This reference has been added to the revised manuscript.

Point 3. Lines 161-168. The mutations in the S protein have been proven to be involved in the systemic spread of the virus, which not always associated to occurrence of FIP (Porter E, Tasker S, Day MJ, et al. Amino acid changes in the spike protein of feline coronavirus correlate with systemic spread of virus from the intestine and not with feline infectious peritonitis. Vet Res. 2014;45(1):49. Published 2014 Apr 25. doi:10.1186/1297-9716-45-49)

Response 3. We agree with this comment and the manuscript has been revised to indicate that the methionine to leucine substitution at position 1058 in the fusion peptide of the FECV spike protein is involved in systemic spread of FCEV from the intestine, rather than with the potential to cause FIP. The study mentioned by the reviewer has also been referenced.

Point 4. Line 168: citation missing.

Response 4. The citation has been added to the revised manuscript.

Point 5. Lines 212-226. The role of metalloproteases secreted by infected monocytes/marcophages in the pathogenesis of FIP should be mentioned.

Response 5. The role of metalloproteases secreted by infected monocytes/macrophages in the pathogenesis of FIP has been mentioned.

Point 6. Line 247. add the following ref:Lorusso A, Decaro N, Schellen P, et al. Gain, preservation, and loss of a group 1a coronavirus accessory glycoprotein. J Virol. 2008;82(20):10312-10317. doi:10.1128/JVI.01031-08.

Response 6. This reference has been added to the revised manuscript.

Point 7. Line 297. Use acronyms.

Response 7. The acronym has been used in the revised manuscript.

Point 8. Line 312. Citation missing. Cite Decaro N, Buonavoglia C. An update on canine coronaviruses: viral evolution and pathobiology. Vet Microbiol. 2008 Dec 10;132(3-4):221-34. doi: 10.1016/j.vetmic.2008.06.007.

Response 8. This reference has been added to the revised manuscript.

Point 9. Line 319. Add the following ref: Alfano F, Dowgier G, Valentino MP, et al. Identification of Pantropic Canine Coronavirus in a Wolf ( Canis lupus italicus) in Italy. J Wildl Dis. 2019;55(2):504-508.doi:10.7589/2018-07-182.

Response 9. This reference has been added to the revised manuscript.

Point 10. Lines 308-314. The authors should discuss that CCoV is not universially recognized as an important canine enteric pathogen, although two independent studies demonstrated that this virus is significantly associated to the occurrence of diarrhoea in dogs (Dowgier G, Lorusso E, Decaro N, et al. A molecular survey for selected viral enteropathogens revealed a limited role of Canine circovirus in the development of canine acute gastroenteritis. Vet Microbiol. 2017;204:54-58. doi:10.1016/j.vetmic.2017.04.007; Duijvestijn M, Mughini-Gras L, Schuurman N, Schijf W, Wagenaar JA, Egberink H. Enteropathogen infections in canine puppies: (Co-)occurrence, clinical relevance and risk factors. Vet Microbiol. 2016;195:115-122. doi:10.1016/j.vetmic.2016.09.006).

Response 10. We agree with this comment and the manuscript has been revised to indicate that CCoV is not universally recognized as an important canine enteric pathogen. The studies mentioned by the reviewer have also been referenced.

Point 11. Lines 361-364. Emerging pathogens have been also associated to CIRD, including canine pneumovirus and Streptococcus equi subsp, zooepidemicus (Decaro N, Mari V, Larocca V, et al. Molecular surveillance of traditional and emerging pathogens associated with canine infectious respiratory disease. Vet Microbiol. 2016;192:21-25. doi:10.1016/j.vetmic.2016.06.009).

Response 11. The manuscript has been revised to include canine pneumovirus and Streptococcus equi subsp. zooepidemicus as additional CIRD pathogens.

Point 12. Lines 395-397. Detection of CRCoV in the canine enteric tract does not mean that the virus has a dual tropism, since most likely this is related not to active replication in the gut but to passive transport from the respiratory tract through ingestion of saliva and respiratory secretions. Please, amend the sentence.

Response 12. We agree with this comment and the manuscript has been revised to indicate that detection of CRCoV in the canine enteric tract may indicate the potential of CRCoV to display a dual tropism, but that it is likely that the detection of CRCoV outside the respiratory tract is a result of passive transport from the respiratory tract through the ingestion of saliva and respiratory secretions.

Point 13. Line 411. Use the acronym.

Response 13. The acronym has been used in the revised manuscript.

Point 14. Paragraph 6.1. No data are reported about epidemiology and clinical features of alpaca coronavirus.

Response 14. Epidemiology and clinical features of both the respiratory and enteric alpaca coronaviruses have been included in the revised manuscript.

Point 15. Paragraphs 5.2 and 6.2. the authors should mention that CRCoV, ECoV and alpaca betacoronavirus are strictly related to BCoV since they descend from the bovine virus or alternatively all these viruses have a common ancenstor, likely a rat betacoronavirus (see Decaro and Lorusso, 2020).

Response 15. We agree with this comment and the manuscript has been revised to mention that CRCoV, ECoV and alpaca betacoronavirus are thought to either descend from the bovine virus or have a common ancestor. In addition, bovine coronavirus has been added to the phylogenetic tree in order to clarify this genetic link.

Point 16. Paragraph 6.3. Nothing about alpaca coronavirus pathology?

Response 16. The manuscript has been revised to include pathological findings for both the respiratory and enteric alpaca coronaviruses.